# Study on Factors That Influence Human Errors: Focused on Cabin Crew

**DOI:** 10.3390/ijerph19095696

**Published:** 2022-05-07

**Authors:** Jiyoung Kim, Myoungjin Yu, Sunghyup Sean Hyun

**Affiliations:** School of Tourism, Hanyang University, 222, Wangsimni-ro, Seongdong-gu, Seoul 04763, Korea; rnrmf.jy@gmail.com (J.K.); rosa7767@hanyang.ac.kr (M.Y.)

**Keywords:** cabin crew, airline, human factors, human errors, job crafting, mental health

## Abstract

This study aims to reduce the possibility of human mistakes and accidents among airline cabin crew by identifying the cause of human errors by focusing on the importance of the causal relationship between human factors and human errors. According to statistical analysis, among the five human factors, physical fatigue, psychological stress, and the complacency of cabin crew had a positive impact on human errors. However, hurrying under time pressure and the distractions caused by external factors do not significantly affect human errors. Human errors have a negative impact on job crafting and mental health. This study analyzed the human factors influencing the cabin crew’s errors and revealed the importance of complacency, which was not covered in previous studies. Finally, the research implications, limitations, and future studies were discussed.

## 1. Introduction

The in-flight service of cabin crew plays a positive role in passenger satisfaction and the airline’s image. In the history of aviation industry development, safe and comfortable journeys are the basics pursued for a high level of airline service [1,2]. Thus, the service quality of cabin crew can be a criterion for passengers’ airline selection [3], and airlines are making a lot of effort to reduce the problem phenomenon, especially when caused by cabin crew mistakes, to improve passenger satisfaction [4].

Cabin crew’s adaptation to various airplane models and excessive work causes physical and mental fatigue, reducing their quality of work, and eventually causes human errors [5,6]. As the required workload increases, there are negative effects on the mind and body of cabin crew, which can reduce the efficiency of their human resource management [7]. According to data from an in-house bulletin board of a major Korean airline, there were many cases of in-flight irregularities and service failures, of which 62% were caused by human errors such as mistakes or carelessness from cabin crew. Therefore, airlines strive to minimize crew human errors through Crew Resource Management (CRM) [8]. However, because CRM has identified stress and fatigue factors as one of the sub-items [8], it is necessary to focus on more detailed research on the human factors necessary for the efficient management and utilization of human resources pursued by CRM.

The positive image of an airline is characterized by airline convenience and in-flight services [9]. For cabin crew to provide high-quality services, efficient human resource management is needed to diagnose cabin crew behavior, prioritize items that require management, and establish and implement the necessary strategies [10,11]. Through this, they can reduce the damage caused by human errors and the loss of manpower from cabin crew injuries [12,13,14]. Despite their importance, there have been no studies, guidelines, or manuals compiled on the types of human errors among airline cabin crew, or their causes. Therefore, this study aims to (1) investigate the effect of human factors on human errors in the conceptual framework, (2) examine the causal relationships among human factors, human errors, job crafting, and mental health through research methods and results, and (3) present theoretical and practical suggestion to minimize in-flight human errors in airlines in the conclusions.

## 2. Conceptual Framework

### 2.1. Human Factors

Human factors have a significant effect on work performance [15]; the risks posed by human factors such as stress, fatigue, and distraction can cause even skilled employees to make mistakes at work [16]. Among human factors, individual psychological factors such as carelessness and oblivion are the most difficult areas to manage, as it is impossible to effectively predict or control the occurrence of mistakes such as attention failure, and it is very difficult to manage them [17]. For example, the results of a marine accident analysis indicate that approximately 75% to 96% of accidents are caused by crew human risk factors. Therefore, research on human risk factors is continuously required [18]. Managing these human factors not only improves work performance and productivity, but also affects the job health and safety of the performer [19,20].

#### 2.1.1. Physical Fatigue

Fatigue is defined as a subjective feeling that exists at a point in a continuum and leads to complete exhaustion, which results from physical, mental, and emotional activities [21]. It is an experience that negatively affects the stability of social, mental, and physical aspects of—and degrades one’s ability to engage with—daily life. It also does not improve with rest [22].

In a study on aircraft maintenance personnel, the majority of respondents said that they made a mistake at work when they felt tired [23]. In addition, according to a study by Camden et al., most serious safety-related accidents occurred at night, and the primary cause of the accident was fatigue [24]. Moreover, aircraft mechanics who worked in shifts said they made more mistakes at work when they felt tired [23]. Similarly, in a study on airline crew, the cabin crew who felt tired were negatively impacted at work, demonstrating poor attention and memory, poor response speed, and poor communication during work [25]. These previous studies led us to propose the following hypothesis.

**Hypothesis** **1** **(H1).** *Physical fatigue of the cabin crew will have a significantly positive (+) effect on human error*.

#### 2.1.2. Psychological Stress

Stress refers to a state in which homeostasis is threatened [26]; all situations or physical environments encountered on a daily basis can cause stress [27]. Stress is sometimes affected by the external environment, but it is an experience of internally perceived negative emotions such as discomfort, anxiety, or pressure felt by an individual [28].

A study by Kerr et al. [29] and another by Bashir and Ismail Ramay [30] showed that stress had a negative effect on work performance. For instance, when pilots are exposed to high stress, they can make hasty decisions due to missing important points or failing to predict the next step [31]. The pilot’s negative psychology and anxious mental state have been shown to cause unsafe behavior [32]. These studies led us to propose the following hypothesis.

**Hypothesis** **2** **(H2).** *Psychological stress among cabin crew will have a significantly positive (+) effect on human errors*.

#### 2.1.3. Hurrying under Time Pressure

Lee stated that hurrying occurs when trying to perform a task in a short time and speeding up performance [33]. Hurry is caused by speeding up behavior, encouraging the omission of some processes, resulting in impatience and embarrassment [34]. This appears as acceleration and omission in time pressures, and goes through the process of minimizing or accelerating cognitive efforts in completing the task [35,36]. Time pressure causes hurry, which leads to strategic behavior with risk taking, as well as physiological activities such as increased heart rate and respiratory rate [37]. Saptari et al. conducted an experiment on workers performing assembly work and found that mistakes increased with increased time-pressure during work [38]. These studies led us to propose the following hypothesis.

**Hypothesis** **3** **(H3).** *Hurrying under time pressure among cabin crew will have a significantly positive (+) effect on human errors*.

#### 2.1.4. Distraction from External Factors

Distraction is the process of reducing or blocking the reception of information by dispersing concentration from areas to which attention should be paid, and is caused by external factors such as visual stimuli, social interaction, and music [39]. Distraction is a digression from the target element which interferes with concentration and immersion [40] and is defined as the intervention of an external object or event that acts to divert attention or sufficiently deprive attention from major areas [41].

Research by Cassidy and MacDonald and Goodell et al. shows that stimuli such as noise cause distraction, resulting in poor work performance or a delay in work completion [42,43]. Distraction can be caused by visual and auditory factors, and can occur even when the flow of work is interrupted or when the equipment is broken [44,45]. When distracted, there is a problem of neglecting the goal of the task performed or the response speed becoming slower than generally expected [46].

Studies that connect distraction and human error are important because the lack of attention-focused abilities can lead to dangerous consequences such as accidents occurring while driving, poor academic performance among students, and in the case of airline performance, the failure to find dangerous objects during baggage inspection [47,48]. Previous studies led us to propose the following hypothesis.

**Hypothesis** **4** **(H4).** *Distraction from external factors in cabin crew will have a significantly positive (+) effect on human errors*.

#### 2.1.5. Complacency

Complacency can be caused by repetition in daily life, which is based on the theory that if the initially perceived experience is repeated, the attitude toward it is strengthened [49]. It can also interpret as excessive confidence, lack of motivation, lack of training, and lack of concentration [50]. Pope and Bogart defined complacency as a dangerous perceived state, and stated that it arises from excessive trust [51,52]. For example, it was revealed that complacency occurred due to the conceit of pilots who were overconfident about automation, and this caused problems such as failure to detect errors, poor work accuracy, and delayed performance [53]. Complacency is one of the main causes of accidents recorded in aviation safety accident reports, along with boredom or decreased attention, and it has been found to cause human errors due to a lowered awareness of risks [54]. These studies led us to propose the following hypothesis.

**Hypothesis** **5** **(H5).** *Complacency among cabin crew will have a significantly positive (+) effect on human errors*.

### 2.2. Human Errors

Rouse and Rouse stated that human errors are common in human behavior and can cause accidents [55]. Human errors include mistakes or failures, which refer to predictive failures, decision errors, inappropriate actions, and instability caused by forgetting or anticipating things during work [56]. In this study, based on the classification of human errors by Reason [47] and Rasmussen [57], we intend to study slips and lapses in which the causes of errors are internal to individuals. These refer to perceptual confusion, interference errors, errors in incorrect arrangements or orders, errors in incorrect timing, errors in omitting procedures or instructions due to memory failure, or errors in repeating pre-performance [57].

The human-centered approach is a method of identifying the cause of human error, focusing on individuals who commit errors and the fact that personal attributes of human beings cause errors [58]. Human-error-causing factor management is associated with individuals, work conditions, and organizational factors. Effective human error management can increase the efficiency of achieving individual or team tasks, situations, and organizational goals [17].

### 2.3. Job Crafting

Job crafting means that employees create and change their job resources to achieve or optimize their performance goals [59]. This has an important impact on employees’ motivation for work, and allows them to develop knowledge and skills, have higher goals, and increase individual growth and satisfaction with their jobs [60,61]. In a study on the correlation between safety performance and job crafting, safety performance was found to have a positive and significant effect on job crafting [62]. It was also inferred that human errors can negatively affect job crafting [62]. These previous studies led us to propose the following hypothesis.

**Hypothesis** **6** **(H6).** *Human errors among cabin crew will have a significantly negative (−) effect on job crafting*.

### 2.4. Mental Health

Mental health refers to an individual’s emotional happiness, ability to live a creative life, and flexibility to cope with the challenges encountered in the process of life, and is defined in terms of satisfaction or happiness [63]. Jahoda suggested positive attitudes toward oneself, personal growth, autonomy, integration, adaptability, environmental mastery—including healthy interpersonal relationships—and true perception of reality as the fundamental categories of mental health [64].

In a mental health study, it was found that mental health problems such as depression or anxiety appear more in people being treated due to work accidents [65]. Through this study, the correlation between errors occurring during work and mental health can be inferred. In addition, in a human error study, it was found that a railway crew’s efforts for mental health management had a positive effect on safe operations [66]. Because efforts to improve mental health are correlated with the reduction of human errors, it can be inferred that human errors can negatively affect mental health [66]. These previous studies led us to propose the following hypothesis.

**Hypothesis** **7** **(H7).** *Human errors in cabin crew will have a significantly negative (−) effect on mental health*.

## 3. Research Methods

### 3.1. Research Models and Hypotheses

The research model shown in Figure 1 was developed based on the conceptual background discussed in Section 2. The conceptual model describes the hypothesized relationships among human factors, human errors, job crafting, and mental health. Based on this theoretical background, this study presents the following seven hypotheses.

**Hypothesis** **1** **(H1).** *Physical fatigue of the cabin crew will have a significantly positive (+) effect on human error*.

**Hypothesis** **2** **(H2).** *Psychological stress among cabin crew will have a significantly positive (+) effect on human errors*.

**Hypothesis** **3** **(H3).** *Hurrying under time pressure among cabin crew will have a significantly positive (+) effect on human errors*.

**Hypothesis** **4** **(H4).** *Distraction from external factors in cabin crew will have a significantly positive (+) effect on human errors*.

**Hypothesis** **5** **(H5).** *Complacency among cabin crew will have a significantly positive (+) effect on human errors*.

**Hypothesis** **6** **(H6).** *Human errors among cabin crew will have a significantly negative (−) effect on job crafting*.

**Hypothesis** **7** **(H7).** *Human errors in cabin crew will have a significantly negative (−) significant effect on mental health*.

### 3.2. Variable Operational Definitions and Survey Items

To develop the measurement tools of this study, definitions of the major variables were presented, and the measurement items were derived. To construct a questionnaire suitable for the study, measurement items suitable for the operational definition of variables were selected.

First, physical fatigue is defined as an experience in which daily life ability deteriorates due to exhaustion, and it is constructed using an equivalent interval scale of four questions derived from previous studies [67,68].

Second, psychological stress is defined as an experience of negative emotions such as internally perceived discomfort, anxiety, and pressure, and is constructed using equivalent interval scales of four questions derived from previous studies [69,70,71].

Third, based on the definition that hurrying under time pressure increases the speed of action for faster performance, it is constructed using the equivalent interval scale of four questions derived from previous studies [72,73].

Fourth, distraction from external factors is defined as the intervention of an external object or event that plays a role in turning or stealing attention from a major area, and four questions derived from previous studies were constructed using an equivalent interval scale [44,45,74,75].

Fifth, complacency is defined as a dangerous recognition state arising from careless or excessive trust, based on unjustified assumptions, and constructed using an equivalent interval scale with four questions derived from previous studies [50,76,77,78].

Sixth, human errors are defined as errors caused by perceptual confusion, misinterpretation, misjudgment, omission of procedures, omission of instructions, and repetitive performance. This section was constructed using an equivalent interval scale, with two questions derived from previous studies [79,80,81].

Seventh, job crafting is defined as creating and changing job resources on their own to achieve job performance goals in the form of active work performance, and this section was constructed using equivalent scales as four questions derived from previous studies [82,83,84].

Mental health is defined as the ability to pursue emotional happiness and live a creative life in a harmonized state of emotional, psychological, and social happiness. It is constructed using an equivalent interval scale with three questions derived from previous studies [84,85].

The equivalent interval scale uses a 5-point Likert scale, where 1 = strongly disagree, 2 = disagree, 3 = neutral, 4 = agree, and 5 = strongly agree. One marks the respondent’s strong negative view, and 5 marks the respondent’s strong positive view.

### 3.3. Collecting Survey Data and Analysis Methods

The survey for this study was conducted on cabin crew from two Korean airlines from 1 December 2020, to 31 March 2021. An online Google survey was conducted using a convenient sampling method. A total of 243 questionnaires were distributed, and statistical analysis was conducted on 239 respondents, excluding four unfaithful people.

To verify the hypotheses, a statistical analysis, frequency analysis, reliability analysis, confirmatory factor analysis, correlation analysis, and structural equation analysis were conducted using IBM SPSS 25 (IBM, New York, NY, USA) and AMOS 25 (IBM, New York, NY, USA).

## 4. Research Results

### 4.1. Demographic Characteristics

Through frequency analysis, the general characteristics of the study subjects were identified, and the results are presented in Table 1. Of the 239 respondents, 34 (14.2%) were men and 205 (85.8%) were women. The age distribution of the respondents was as follows: 48 people (20.1.%) in their 20s, 127 (53.1%) in their 30s, 40 (16.7%) in their 40s, and 24 (10%) in their 50s. There were 137 unmarried (57.3%) and 102 married (42.7%) people. 47 people (19.7%) had a college degree, 165 (69%) had a university degree, 10 (4.2%) were graduate students, and 17 people (7.1%) had a graduate degree.

In terms of the distribution of work grades, 87 (36.4%) were flight attendants, 93 (38.9%) were assistant pursers, 42 (17.6%) were pursers, 14 (5.9%) were senior pursers, and 3 (1.3%) were chief pursers. As for the period of employment, 40 (16.7%) had worked for less than five years, 86 people (36.0%) between 5 and 9 years, 52 people (21.8%) between 10 and 14 years, and 61 people (25.5%) for over 15 years. As for their positions in the workplace, 45 people (18.8%) were managers, 65 people (27.2%) were upper-class galley, 55 people (23%) were upper-class aisles, 20 people (8.4%) were economy-class galley, and 54 people (22.6%) were economy-class aisles.

The annual salary of the five people (2.1%) was less than KRW 30 million. 45 people (18.8%) earned between KRW 30 to 40 million, 39 people (16.3%) between KRW 40 to 50 million, 47 people (19.7%) from KRW 50 to less than 60 million, and 103 people (43.1%) earned over KRW 60 million. 205 people (85.8%) in the survey worked for Korean Air and 34 people (14.2%) worked for Asiana Airline.

### 4.2. Confirmative Factor Analysis and Reliability Analysis

This study conducted a confirmatory factor analysis (CFA) to verify the validity of the composition of observed variables for latent variables before verifying the structural relationship among cabin crew physical fatigue, psychological stress, hurrying under time pressure, distraction from external factors, and complacency. The CFA model used for the validation is shown in Figure 2.

The results of the CFA for the measurement model are presented in Table 2 and Table 3. A factor loading value of all items of 0.50 or higher was statistically significant. The model fit was determined as CFI = 0.947, TLI = 0.938, RMSEA = 0.057. If the CFI was 0.90 or higher, TLI was 0.90 or higher, and RMSEA was 0.08 or less. The measurement model fit was considered good and valid [86].

As a result of confirming the internal consistency of items, within the configured factors, through Cronbach’s α, physical fatigue was 0.895, psychological stress was 0.916, hurrying under time pressure was 0.727, distraction from external factors was 0.846, complacency was 0.899, human error was 0.953, job crafting was 0.889, and mental health was 0.837. Cronbach’s α for all factors was found to be 0.70 or higher, and the reliability of the measurement model was considered effective.

Before proceeding with the structural model analysis, the consistency and convergence of the observed variables constituting the latent variable, whether the similarity among the latent variables was excessively high, and whether there was an independent concept through convergence validity verification was judged.

Table 3 shows the results of the convergent validity verification. Convergence feasibility can be determined based on the composite reliability (CR) and average variation extracted (AVE). In general, convergence feasibility is determined as acceptable when the concept of reliability is higher than 0.70, and the average variance extraction is higher than 0.50 [86]. As a result of the statistical analysis of this study, both the CR and AVE of all variables met the reference values, and it was determined that there was no problem with the convergence validity.

### 4.3. Correlation and Discriminant Validity Analysis

The results of the correlation coefficients among the potential variables of the measurement model are presented in Table 4. There was a significant positive correlation among independent variables, with physical fatigue, psychological stress, hurrying under time pressure, distraction from external factors, complacency, and human errors as parameters. However, there was a significant negative correlation among human errors as a parameter and the dependent variables of job crafting and mental health.

The correlation among potential variables showed significant results overall, and the correlation coefficient between complacency and human errors was the highest at 0.864.

The discriminant validity verification proposed by Bagozzi and Yi was conducted to determine whether such a high correlation hinders the discriminant validity of the measurement model [87]. The correlation between complacency and human errors was the highest, and the chi-square values of the merged model between complacency and human errors and the original model separating complacency and human errors were compared. The results of this comparison are presented in Table 5.

The chi-square value of the original model separating complacency and human errors was 618.485, and the degrees of freedom was 349. The chi-square value of the model that integrated complacency and human errors into a single factor was 724.768, and the degree of freedom was 356. In other words, the chi-square value differed by 106.283, and the degrees of freedom differed by seven. The chi-square value difference between the two models is higher than the threshold of 14.067. Therefore, the model that distinguishes between complacency and human errors is the better one. CFI and TLI, which have high and good suitability, were found to be higher, and RMSEA, which has a good—if low—suitability, was found to be better in the original model that separated complacency and human errors.

As a result, it has been verified that the original model separating complacency and human errors is a better model, thereby determining that the measurement model has discriminant validity.

### 4.4. Structural Equation Model Analysis

This study aims to investigate the effect of five human factors on human errors: physical fatigue, psychological stress, hurrying under time pressure, distraction from external factors, and complacency. In addition, it aims to identify the effects of human errors on job crafting and mental health.

To achieve the purpose of this study, a structural equation model was constructed, as shown in Figure 3.

Table 6 shows the confirmation results determining the suitability of the structural equation model configured for this study.

The major goodness-of-fit index was CFI = 0.938, TLI = 0.930, and RMSEA = 0.061. CFI and TLI were above 0.90, and RMSEA was below 0.08, satisfying the goodness-of-fit index criteria for structural equation models. Therefore, the goodness-of-fit of the structural equation model was determined to be acceptable for this study.

Table 7 shows the results of verifying the direct effect between variables in the SEM, by confirming the path coefficient and its significance in the structural model.

The path from physical fatigue to human errors showed positive (+) significant results (β = 0.137, *p* < 0.05). It can be judged that the higher physical fatigue, the higher the number of human errors. The route from psychological stress to human errors also showed positive (+) significant results (β = 0.236, *p* < 0.01). It can be determined that higher psychological stress results in higher human errors. The route from hurrying under time pressure and distraction due to external factors to human errors was found to be insignificant, but the route from complacency to human errors showed significantly positive (+) results (β = 0.601, *p* < 0.001). It can be judged that the higher the complacency, the higher the human errors. It was found that the β value (0.601) was the highest among the paths from human factors to human errors.

The path from human errors to job crafting showed significantly negative (−) results (β = −0.574, *p* < 0.001). It can be determined that more frequent human errors resulted in lower job crafting. In addition, the route from human errors to mental health also showed significantly negative (−) results (β = −0.844, *p* < 0.001). It can be judged that more frequent human errors resulted in a lower level of mental health.

Therefore, hypotheses 1, 2, 5, 6, and 7 of this study are supported, while hypotheses 3 and 4 are not supported, as shown in Figure 4.

## 5. Conclusions

### 5.1. Discussions

This study conducted an empirical analysis to review the literature on five human factors affecting human errors among cabin crew, and to determine its effects on job crafting and mental health. Through previous studies, the five human factors affecting human error were found to be physical fatigue, psychological stress, hurrying under time pressure, distraction from external factors, and complacency. Additionally, the factors had an impact on job crafting and mental health through human errors.

Among five human factors, physical fatigue, psychological stress, and complacency were found to have serious effect on the crew’s human error. This means that, as shown in previous studies by Armentrout et al. and Tvaryanas and MacPherson, fatigue-rich conditions lead to decreased attention and situational awareness, which can lead to increased human errors [88,89]. In addition, high stress levels were found to cause unstable behavior [32] or cause frequent errors and mistakes [29]. This means that managing cabin crews’ physical fatigue and mental stress can reduce human errors. In particular, complacency was found to have the highest influence on inducing human errors as the factor that lowers awareness of risk [54]. This shows that cabin crews who repeatedly experience abnormal situations or problems during flights, or are not motivated to perform their duties, are more likely to cause human errors.

Cabin crew who frequently made human errors showed poor job crafting. As a result, their self-directed or active work performance decreased. It was also found that human errors negatively affected mental health, causing emotional anxiety and making it impossible to cope with the problems that occur during work.

### 5.2. Implications

The academic implications of this study show that it is meaningful that the causal relationship between the crew’s human factors that affect their human errors have been revealed. The human error of a cabin crew during flying is directly related to customer satisfaction, and is a very important part of the airline service. However, research on the causes of such human errors has not been conducted yet. This study contributes by presenting a theoretical basis for research on the causes and effects of human errors in cabin crew.

The practical implications of this study are as follows: As the cause of human errors among cabin crew, the human factor of complacency was found to be the most influential. It contributed to expanding the scope of human resource management items and associations. Previous studies on human errors have mainly focused on the fatigue or stress of human factors. However, in this study, complacency has a greater influence as the cause of human error, revealing the need for more nuanced research on complacency. Therefore, through this study, it was possible to understand the structure of human factors affecting human errors, and to set guidelines to reduce the possibility of causing mistakes and problems among cabin crew.

### 5.3. Limitations and Recommendations for Future Research

There is a limit to generalizing the research results of this study because it only focuses on cabin crew working in Korea as its sample targets. The schedule allocation methods, standards, routes, and types of work in airlines vary from country to country, which can lead to differences in the workplace experiences of cabin crew. To verify this study in other geographical contexts, it will be necessary to distribute the research subjects evenly around the world when research related to in-flight human factors is conducted in the future.

This study also does not focus on the cause of human errors, which may differ depending on personal characteristics such as working years and positions. The difference could not be measured in this study, as it was outside its scope. If the study is conducted by further subdividing the subject’s personal characteristics, their specific degrees of involvement in the cause of human factors may appear differently.

## Figures and Tables

**Figure 1 ijerph-19-05696-f001:**
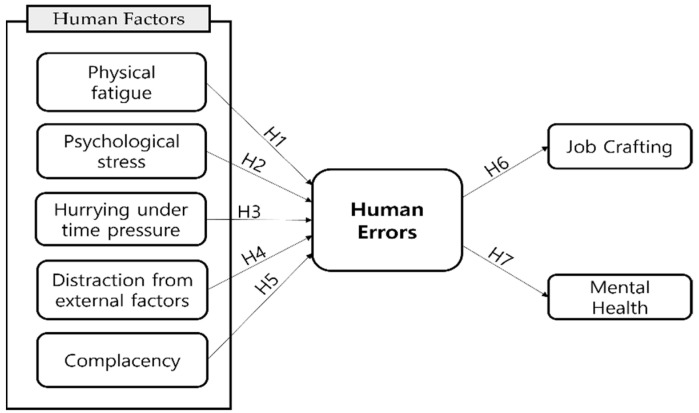
Research model.

**Figure 2 ijerph-19-05696-f002:**
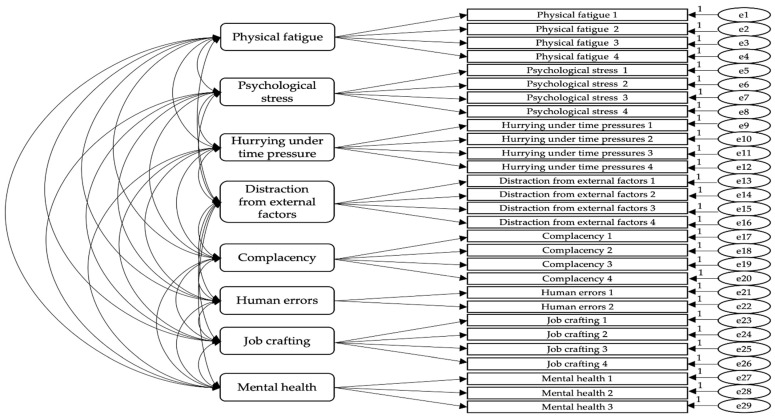
Confirmative Factor Analysis Model.

**Figure 3 ijerph-19-05696-f003:**
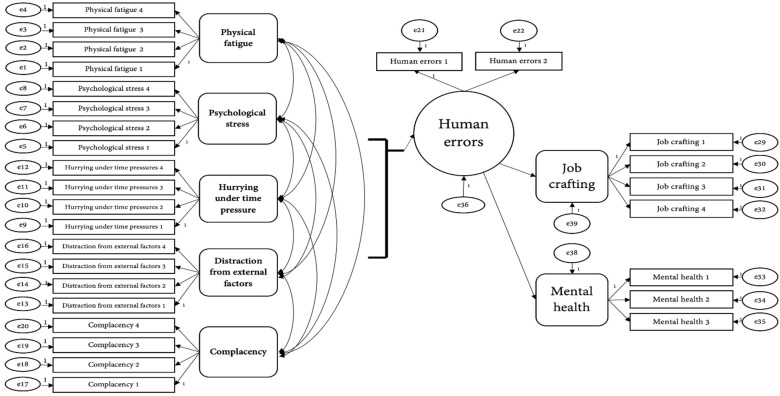
Structural equation model.

**Figure 4 ijerph-19-05696-f004:**
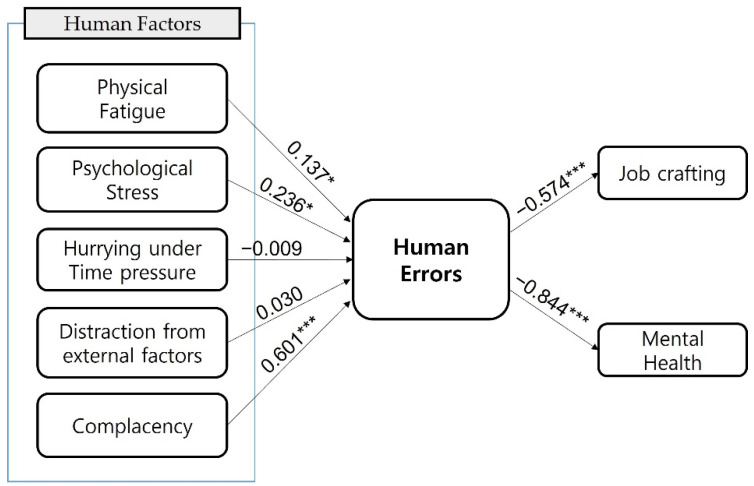
Structural equation model analysis. * *p* < 0.05, *** *p* < 0.001.

**Table 1 ijerph-19-05696-t001:** Characteristics of the Subject Targets.

Variables	Index	Frequency (n)	Percent (%)	Mean (SD)
Gender	Male	34	14.2	
	Female	205	85.8	
Age	20 s	48	20.1	35.91 (7.61)
	30 s	127	53.1	
	40 s	40	16.7	
	50 s	24	10.0	
Marital status	Single	137	57.3	
	Married	102	42.7	
Education	College degree	47	19.7	
	University degree	165	69.0	
	In graduate school	10	4.2	
	Graduate degree	17	7.1	
Working grade	Flight attendant	87	36.4	
	Assistant Purser	93	38.9	
	Purser	42	17.6	
	Senior Purser	14	5.9	
	Chief Purser	3	1.3	
Period of employment	Under 5 years	40	16.7	11.30 (8.03)
	5–10 years under	86	36.0	
	10–15 years under	52	21.8	
	Over 15 years	61	25.5	
Position at workplace	Manager	45	18.8	
	Upper-class galley	65	27.2	
	Upper-class aisle	55	23.0	
	Economy-class galley	20	8.4	
	Economy-class aisle	54	22.6	
Airline	Korean air	205	85.8	
	Asiana airlines	34	14.2	
Annual income	KRW under 30 million	74	37.0	
	KRW 30–40 million	55	27.5	
	KRW 40–50 million	71	35.5	
	KRW 50–60 million			
	Over KRW 60 million	105	52.5	
Total		239	100.0	

**Table 2 ijerph-19-05696-t002:** Results of CFA and Reliability Analysis.

Factors	Measurement Items	M (SD)	Loading	Cronbach’s α
Physical fatigue	I often feel that I want to rest during work.	3.62 (0.79)	0.734	0.895
	I am often sleepy due to jet lag or lack of sleep during work.	3.58 (0.85)	0.838	
	I often feel that it is difficult to continue my work due to exhaustion.	3.32 (0.86)	0.899	
	I often feel that my physical strength is not enough during work.	3.28 (0.85)	0.831	
Psychological stress	I often feel confused or unable to control my emotions during work.	3.13 (1.10)	0.887	0.916
	I often feel frustrated about uncontrollable situations, such as being assigned unwanted tasks during work.	3.43 (1.10)	0.852	
	I have often experienced depression, anxiety, and emotional exhaustion during work.	3.34 (1.09)	0.858	
	I often get offended and angry by others while working.	3.70 (0.94)	0.827	
Hurrying under time pressure	I often feel that I am not given enough time to perform my work.	4.16 (0.74)	0.543	0.727
	I often feel that I have to speed up my work compared to the other cabin crew.	3.93 (0.88)	0.625	
	I often rush to get things done quickly during work.	4.00 (0.81)	0.753	
	I often feel I have to hurry because I have received too many requests from passengers or colleagues during work.	4.08 (0.76)	0.640	
Distraction from external factors	I often get distracted by the sound of conversation, in-flight noises, and the other person’s attempts to talk.	3.31 (0.98)	0.846	0.846
	I am often distracted by inspections by external organizations or interference from superiors during work.	3.63 (1.04)	0.742	
	I often get distracted when the flow of work is interrupted, such as experiencing turbulence.	2.97 (1.06)	0.853	
	I am often distracted by visual factors such as dirty galleys or dark cabin environments.	3.11 (1.07)	0.698	
Complacency	I often don’t get involved even though I know that the situation is wrong, such as a colleague’s mistakes at work.	2.95 (0.99)	0.844	0.899
	In the case of joint responsibility at work, I often omit the task because I think my colleague would have performed it instead.	2.90 (1.08)	0.927	
	I often think there will be no major problems such as hijacking or emergency landing during the flight.	3.47 (0.93)	0.739	
	I often have no motivation to actively perform my work due to standardized evaluation systems, reduced promotion opportunities, or omission of upper class work.	3.23 (0.92)	0.816	
Human errors	I often experience miscommunicating instructions due to information confusion, misjudgment, and misunderstanding during flight.	3.23 (1.02)	0.967	0.953
	I often experience forgetting requests from colleagues or passengers, or making mistakes in repeating what I have already done during the flight.	3.39 (0.94)	0.899	
Job crafting	I tend to try to learn new things during work.	3.30 (0.94)	0.769	0.889
	I tend to decide on how to work by myself.	3.42 (0.92)	0.749	
	I tend to understand my work and find a way to perform better.	3.51 (0.93)	0.893	
	I tend to try to develop myself professionally during work.	3.42 (0.89)	0.862	
Mental health	I tend to be sensitive and anxious when I work.	2.60 (1.13)	0.708	0.837
	I often have little interest in work.	2.86 (1.19)	0.858	
	I tend to think that I don’t have much hope in the future for work.	2.85 (1.19)	0.814	
	χ^2^ = 618.485, df = 349, *p* < 0.001, CFI = 0.947, TLI = 0.938, RMSEA = 0.057			

**Table 3 ijerph-19-05696-t003:** Convergence Feasibility Validation.

Variations	Composite Reliability (CR)	Average Variance Extracted (AVE)
Physical fatigue	0.927	0.761
Psychological stress	0.909	0.714
Hurrying under time pressure	0.815	0.527
Distraction from external factors	0.839	0.567
Complacency	0.908	0.713
Human errors	0.937	0.881
Job crafting	0.906	0.708
Mental health	0.929	0.623

**Table 4 ijerph-19-05696-t004:** Verification of Relationship between each Potential Variables.

Variations	1	2	3	4	5	6	7	8
1. Physical fatigue	1							
2. Psychological stress	0.706 ***	1						
3. Hurrying under time pressure	0.438 ***	0.614 ***	1					
4. Distraction from external factors	0.728 **	0.814 ***	0.572 ***	1				
5. Complacency	0.657 ***	0.720 ***	0.386 ***	0.772 ***	1			
6. Human errors	0.687 ***	0.743 ***	0.413 ***	0.750 ***	0.864 ***	1		
7. Job crafting	−0.417 ***	−0.518 ***	−0.283 ***	−0.448 ***	−0.549 ***	−0.529 ***	1	
8. Mental health	−0.689 ***	−0.797 ***	−0.455 ***	−0.737 **	−0.761 ***	−0.801 ***	624 ***	1

Note: diagonal—(normalized); diagonal down—correlation coefficient. ** *p* < 0.01, *** *p* < 0.001.

**Table 5 ijerph-19-05696-t005:** Discriminative validity reasoning based on Bagozzi and Yi.

Model	χ^2^	df	Δχ^2^	Δdf	CFI	TLI	RMSEA
Original	618.485	349			0.947	0.938	0.057
Merge	724.768	356	106.283	7	0.926	0.915	0.066

**Table 6 ijerph-19-05696-t006:** Validity of the structural equation model.

χ^2^	Df	*p*	CFI	TLI	RMSEA
673.772	360	<0.001	0.938	0.930	0.061

**Table 7 ijerph-19-05696-t007:** Structural model path analysis.

Hypothesis.	Relationship			B	SE	β	C.R.	p	Decision
H1	Physical fatigue	→	Human errors	0.228	0.101	0.137	2.262 *	0.024	Supported
H2	Psychological stress	→	Human errors	0.233	0.082	0.236	2.849 **	0.004	Supported
H3	Hurrying under time pressure	→	Human errors	−0.021	0.136	−0.009	−0.151	0.880	Not supported
H4	Distraction from external factors	→	Human errors	0.039	0.125	0.030	0.313	0.754	Not supported
H5	Complacency	→	Human errors	0.690	0.084	0.601	8.210 ***	0.000	Supported
H6	Human errors	→	Job crafting	−0.431	0.051	−0.574	−8.442 ***	0.000	Supported
H7	Human errors	→	Mental health	−0.685	0.062	−0.844	−11.078 ***	0.000	Supported

* *p* < 0.05, ** *p* < 0.01, *** *p* < 0.001.

## Data Availability

Data sharing not applicable.

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
