# Peer review of "Study on Factors That Influence Human Errors: Focused on Cabin Crew"

_ijerph, 2022, doi:10.3390/ijerph19095696_

Round 1

Reviewer 1 Report

Dear authors,

Congratulations for your interesting and innovative paper. In order to improve it I would suggest to reformulate your keywords. You write a lot of them. Please be more concise here. 

Please add the structure of the paper in the introduction section.

Also add a section concerning Conclusions. You can for instance integrate section 5, 6, 7 and the same section of "Conclusions". 

good work in the next steps.

Author Response

Dear. Reviewer

We are very honored to receive valuable comments related to this study from expert reviewers. We think it was a great support in improving the quality of this study.

We are sending you a detailed response to the reviewers' comments in the attached file. The title of the paper is “Study on factors that influence human errors: Focused on cabin crew”. Please refer to it.

If you have any questions about this study, please feel free to contact us. We look forward to hearing from you soon.

Thank you very much.

Sincerely,

Myoungjin Yu

Reviewer 2 Report

The research is interesting. Presented correctly. I have no methodological comments. The "discussion" needs to be improved.

Author Response

Dear. Reviewer,

We are very honored to receive valuable comments related to this study from expert reviewers. We think it was a great support in improving the quality of this study.

We are sending you a detailed response to the reviewers' comments in the attached file. The title of the paper is “Study on factors that influence human errors: Focused on cabin crew”. Please refer to it.

If you have any questions about this study, please feel free to contact us. We look forward to hearing from you soon.

Thank you very much.

Sincerely,

Myoungjin Yu

Reviewer 3 Report

Dear Authors,
Your article is interesting, thoughtful and well written. A possible improvement could refer to the suggestions of future research related to the development of the model presented by the authors, as the proposed verification in other geographical contexts is quite obvious.

Author Response

(The authors gave the same response as above.)
